# Broadly Applicable Control Approaches Improve Accuracy of ChIP-Seq Data

**DOI:** 10.3390/ijms24119271

**Published:** 2023-05-25

**Authors:** Meghan V. Petrie, Yiwei He, Yan Gan, Andrew Zachary Ostrow, Oscar M. Aparicio

**Affiliations:** Molecular and Computational Biology Section, University of Southern California, Los Angeles, CA 90089, USA; mpetrie@usc.edu (M.V.P.); yiweihe@usc.edu (Y.H.); yangan@usc.edu (Y.G.); zacostrow@gmail.com (A.Z.O.)

**Keywords:** chromatin immunoprecipitation, controls, DNA-binding protein, genome, replication origins

## Abstract

Chromatin ImmunoPrecipitation (ChIP) is a widely used method for the analysis of protein–DNA interactions in vivo; however, ChIP has pitfalls, particularly false-positive signal enrichment that permeates the data. We have developed a new approach to control for non-specific enrichment in ChIP that involves the expression of a non-genome-binding protein targeted in the IP alongside the experimental target protein due to the sharing of epitope tags. ChIP of the protein provides a “sensor” for non-specific enrichment that can be used for the normalization of the experimental data, thereby correcting for non-specific signals and improving data quality as validated against known binding sites for several proteins that we tested, including Fkh1, Orc1, Mcm4, and Sir2. We also tested a DNA-binding mutant approach and showed that, when feasible, ChIP of a site-specific DNA-binding mutant of the target protein is likely an ideal control. These methods vastly improve our ChIP-seq results in *S. cerevisiae* and should be applicable in other systems.

## 1. Introduction

Protein–DNA interactions control all aspects of genome function, from the histones that compactly organize and regulate access to DNA to trans-acting factors binding DNA to regulate cis-elements and also to factors interacting indirectly with DNA through chromatin. Chromatin immunoprecipitation analyzed using DNA sequencing (ChIP-seq) is a widely used and relied upon method for the determination of the genomic binding sites of chromatin-associated proteins (Figure 1A) [1,2,3]. The efficacy of ChIP relies on the antibody-mediated immunoprecipitation (IP) of the intended target protein. Many available antibodies are ineffective or unavailable in sufficient quantities, so a common approach is the use of well-characterized monoclonal antibodies against commonly used peptide epitopes such as HA, MYC, and FLAG, which may be encoded into target proteins, usually without disruption to the native function. In addition to the key challenge of detecting bona fide interactions (false-negatives), one of the persistent problems associated with ChIP-seq is false-positive signals, which can be pervasive [4,5,6]. While several factors might contribute to false positives, one genomic feature that has been correlated with false-positive ChIP signals is the level of transcription, with highly transcribed regions frequently exhibiting signals across unrelated experiments, this phenomenon being referred to as hyperChIPability [5].

Because the enrichment for hyperChIPable sequences is associated with the IP step, neither an input chromatin sample nor immunoprecipitation in the absence of antibody are adequate controls that enable the subtraction of this false signal. Furthermore, ChIP with an antibody versus epitope tag in a strain lacking an epitope-tagged protein exhibited variability in the detection of hyperChIPable signals [5]. In this previous study, a more useful control to identify hyperChIPable signals for subtraction from total signals appeared to be ChIP of cells expressing an unrelated, non-DNA-binding protein, nuclear-expressed green fluorescent protein, the enrichment of which correlated with locus expression levels. These results suggest that a combination of factors, including the nature and expression of the target epitope/protein as well as the characteristics of specific antibodies, influence ChIP-enrichment of false binding loci. To solve this problem, we aimed to develop a universal approach based on epitope tag sharing by the protein of interest and a control protein, which is not expected to bind the genome specifically. Hence, ChIP in parallel with a strain expressing only the control protein would provide data on non-specific enrichment for background subtraction using normalization. We also tested an approach utilizing a specific DNA-binding mutant of the protein of interest. Both methods vastly improve the accuracy of our ChIP-seq datasets, particularly in removing likely false positives.

## 2. Results

### 2.1. Expression of an Epitope-Tagged Protein as Normalization Control

In the absence of more definitive controls, the examination of highly expressed and other hyperChiPable loci is fraught with ambiguity. More broadly, the variability in ChIP enrichment, due to multiple potential factors indicated above, emphasizes the continuing need for better control approaches. We devised a scheme to generate ChIP controls that allow for the identification of false-positive signals while simultaneously providing a control for internal and/or external quantitative normalizations. The scheme was designed for the analysis of epitope-tagged proteins, focusing here on the use of the common MYC, HA, and FLAG epitopes. We generated constructs that express bacterial LexA protein with a C-terminal fusion of one of the three epitopes and contain lexA operator sites (lexA-Op) to serve as sequence-specific binding site(s) for the LexA-epitope-tagged protein (Figure 1B). These constructs, expressing LexA-3xHA, LexA-13xMYC, and LexA-3xFLAG, were stably integrated into an otherwise untagged, wild-type yeast strain and referred to as HOP, MOP, and FLOP, respectively. These control strains would be modified by the introduction of the same epitope-tag onto the gene encoding the protein of interest. The expressions of tagged proteins were confirmed using immunoblot analysis (Appendix A).

We envisioned that a strain bearing the MOP control construct would be analyzed using ChIP alongside the identical strain also expressing “your-favorite-gene” (YFG) tagged with MYC (*YFG-MYC* (MOP)), with the expectation that differential analysis will reveal bona fide ChIP-enrichment for the target protein (Figure 1C). In addition, tagged-LexA binding enrichment observed at the lexA-Op sites was anticipated to serve as a standard for quantitative normalization directly between experimental replicates or as an internal standard versus other queried sites in the genome. As such, the control locus also should permit the evaluation and potential correction of technical variance in the IP between individual samples and replicates.

To begin to evaluate the utility of the approach, we performed ChIP-seq of untagged strains subject to IP with anti-MYC, anti-FLAG, or anti-HA. Triplicates of each ChIP sample were produced and, following mapping to the genome and binning, compared to each other in two-dimensional scatter plots to confirm a high degree of correlation between replicates (Appendix A). Triplicates were read-count normalized and averaged. For comparisons, averaged datasets were scale-normalized using the ChIP signal at the 20th percentile expected to represent unenriched, background signals across all samples.

We analyzed ChIP signals at the lexA binding sites within the integrated control constructs. Three different constructs were tested: LexA-13xMYC and LexA-3xHA were expressed from the *ADH1* promoter, and these vectors contained one lexA binding array with four semi-palindromic binding sites for the LexA dimer (Figure 1B); MOP and FLOP were integrated next to *ADE2* on chr XV, while HOP was integrated next to *TRP1* on chromosome IV (Figure 1B and Figure 2A). LexA-3xFLAG was expressed from the stronger *TEF1* promoter, and the integrating vector contained two separate lexA binding arrays, one with two and one with four lexA binding sequences (Figure 1B). We plotted data for the MOP, FLOP, and HOP loci in the control ChIPs showing binding at these loci, specifically in the strain expressing the corresponding epitope-tagged LexA (Figure 2A); data for individual replicates show good reproducibility (Appendix A). The FLOP construct that contains two lexA binding arrays additionally provides a useful demonstration of the resolution of the data, showing a clear separation between the two data peaks corresponding to the two lexA arrays, separated by ~1.6kb (Figure 2A). These control loci can be used for normalization across sample replicates or internal standardization. However, to compare independent ChIP samples across different IPs, with and without target protein expression, we find that a normalization approach based on the data distributions, such as the 20th percentile background we have applied, is more reliable and established, and we have used that method throughout this paper. Hereon, we focus on the utility of MOP/FLOP/HOP expression as normalization controls for background subtraction.

### 2.2. Non-Specific Signals Are Pervasive in ChIP

We examined averaged ChIP signals for 500 bp bins of consecutive, non-overlapping sequences for the whole genome using quartile boxplot analysis. An examination of the ChIP signal distribution across the genome amongst the different control IPs in untagged strains shows a wide range of signals and numerous outliers (3–8% of all bins), strongly suggesting false enrichment (Figure 2B). In the strains expressing a MOP, FLOP, or HOP internal control construct, the ChIP signal distributions showed significant differences in comparison with the corresponding IP in the untagged strain (Figure 2B, Appendix A). Moreover, two-dimensional scatter plots show that sequences are differentially enriched depending on the IP; for example, the sequences with the greatest enrichment in one IP usually did not show the greatest enrichment in different IPs (Figure 2C). Overall, the results show that significant, non-specific signals are differentially enriched in these ChIPs, and that the sequence composition of this enrichment is highly dependent on the specific antibody, as well as the presence of a target protein (e.g., MOP, FLOP, HOP).

To further examine the nature of these non-specifically IP’d sequences, we analyzed the ChIP signal distribution of 238 loci previously defined as hyperChIPable [5], as well as *tRNA* genes, which are highly expressed, a characteristic that has been correlated with non-specific ChIP enrichment. In comparison with the overall signal distributions from each IP, signals for hyperChIPable and *tRNA* genes were significantly under-enriched in untagged strains, except for hyperChIPable in IP-FLAG, and over-enriched in the MOP, FLOP, and HOP strains, except for hyperChIPable in IP-MYC (Figure 2B, Appendix A). While this lack of significant enrichment for most of these controls might seem contrary to expectations, we find that much variability exists in those sequences that are non-specifically enriched in different IPs. Importantly, as shown below, enrichments in the HOP, MOP, and FLOP strains more effectively represent non-specific enrichment present in the experimental IPs, while the lack of enrichment of these sequences in the Untag control IPs undermines its usefulness as a control.

### 2.3. Ratio Normalization by Epitope-Tagged Control Refines Data Quality for Target Proteins

Having established IPs from untagged and control strains as containing non-specifically enriched sequences, we moved to the analysis of a target protein. We chose Fkh1 due to our continuing interest in maximizing the accuracy of our data and, hence, our understanding of the chromatin dynamics of this protein that has been implicated in the regulation of replication, transcription, and recombination during mating-type switching in yeast [7]. In this regard, existing knowledge of Fkh1 binding at specific replication origins, gene promoters, and the recombination enhancer provides bona fide loci against which to validate our new approach. Additionally, we had previously analyzed Fkh1 using ChIP-chip and reported its association with multiple loci, including Pol III-transcribed loci such as *tRNAs*, and considered these previous results ripe for re-evaluation as potential hyperChIPable artifacts [8].

Fkh1-MYC was expressed from its endogenous locus in an otherwise untagged strain and in the MOP control strain; similarly, Fkh1-FLAG was expressed in untagged and the FLOP strain. These strains were subjected to ChIP-seq alongside the corresponding control strains described above. Analysis of the replicates showed high reproducibility (Appendix A), and replicates were counts-normalized and averaged, after which datasets were scale-normalized at the 20th percentile of the distributions. To apply the normalization controls against their respective experimental sample, the Fkh1-MYC and Fkh1-FLAG experimental results were divided by the corresponding control datasets, which we refer to as ratio normalization (RN) (Figure 1C). We created distribution boxplots (500 bp) and heatmaps (5 kb) of the regions surrounding chromosome features, such as “Forkhead-activated” replication origins and “*CLB2*-cluster” genes, where Fkh1 is known to function and a subset of which have previously been shown to bind Fkh1 [8,9,10]. We also analyzed ChIP signals at *tRNA*s, where we previously detected the binding of Fkh1, though no function has been ascribed, and at hyperChIPable loci, we expected to represent false-positive artifacts of ChIP.

We begin with an analysis of Fkh1-MYC, which we analyzed previously using ChiP-chip [8]. Boxplots and heatmaps show the enrichment of Fkh1-MYC at hyperChIPable and *tRNA* loci in both otherwise untagged and MOP strains (Figure 3). 

Notably, this enrichment was also present in the MOP strain but not in the untagged strain (Figure 2B and Figure 3B, Appendix A), such that RN effectively eliminates the enrichment of these dubiously enriched loci in the MOP strain background (*FKH1-MYC* MOP-RN) but not in the untagged strain background (*FKH1-MYC* Untag-RN) (Figure 3A,B, Appendix A). ChIP signals at *tRNA* genes were not significantly enriched in the untagged and MOP control strains, but RN resulted in the significant enrichment of *tRNA* genes in the untagged set (UntagRN), as in the previous study, but not in the MOP-controlled set (MOP-RN) (Figure 3A,B, Appendix A). In contrast, an examination of the sequences expected to bind Fkh1 yielded different results, showing significant enrichments of Fkh-activated origins in both Fkh1-MYC strains that were not eliminated but enhanced by RN (Figure 3A,B, Appendix A), while *CLB2*-cluster genes showed enrichment in the heatmaps after RN but did not meet statistical significance for the *CLB2*-cluster genes as a group (Figure 3A,B, Appendix A).

Identical analysis of the Fkh1-FLAG data yielded similar overall results with hyperChIPable signals being enriched in the Fkh1-FLAG datasets and being eliminated by RN, in this case with both corresponding control datasets, whereas *tRNAs* remained significantly enriched only in the Fkh1-FLAG UntagRN (Figure 3A,B, Appendix A). As above, RN did not eliminate Fkh1-FLAG signal enrichment from the expected binding loci, including Fkh-activated origins and *CLB2*-cluster genes, and again, signal enrichment at Fkh-activated genes was highly significant but not at *CLB2*-cluster genes (Figure 3A,B, Appendix A). Together with the MYC epitope results, these data suggest that the application of our control paradigm significantly improves the quality of ChIP data by eliminating likely false positives while retaining likely true positives. In this regard, it is interesting that both Fkh1-MYC UntagRN and Fkh1-FLAG UntagRN showed significant enrichment for *tRNA* genes reproducing our previously published ChIP-chip results, whereas these significant enrichments were eliminated by MOP- and FLOP-RN, suggesting that Fkh1 binding at *tRNA* genes was a ChIP artifact [8]. We observed similar results for *snoRNAs* that were another category of features strongly enriched in previous ChIP-chip analysis of Fkh1-MYC with an untagged strain as a subtraction control. As with *tRNAs*, *snoRNAs* showed enrichment in Fkh1-MYC Untag-RN in our current experiments, essentially reproducing the previous results; however, MOP-RN showed no enrichment for *snoRNAs*, suggesting that it too was an artifact of ChIP (Appendix A).

### 2.4. Fkh1 Analysis in G1 Phase to Test Known Enrichment at Replication Origins

We created chromosome plots to view the data along chromosomes III-L and IX, which contain several Fkh1-binding loci of interest indicated on the plots (Appendix A). In these data from unsynchronized cultures, Fkh-activated replication origins and *CLB2*-cluster genes showed relatively minor peaks of enrichment in comparison to the recombination enhancer (RE), which contains numerous Fkh1 consensus binding sequences. We generated data that we could subject to a more stringent analysis by synchronizing the *FKH1-MYC*(MOP) and *FKH1-FLAG*(FLOP) and MOP and FLOP control strains in G1 phase with ⍺-factor, when Fkh1 binds to a subset of replication origins, presenting an expectation of more robust detection than in unsynchronized cells. Experimental replicates show a high correlation (Appendix A), and an analysis of the control IPs shows that different sequences are non-specifically enriched in the ChIP of G1 versus asynchronous cells (Appendix A). We generated chromosome plots to visualize the uncontrolled and normalized data along chromosomes III-L and IX, which contain several Fkh1-binding loci of interest indicated on the plots (Figure 4A). The plots also indicate peaks called by MACS, which are summarized in the chart (Figure 4B). As expected, the G1 data show more called peaks that overlap with the Fkh-activated replication origins than in the unsynchronized data. Peak calling by MACS also independently verifies the specificity of the results and enhancement of the data through our control approach. Peak calling is notoriously challenging and imperfect but presents an orthogonal analysis of the data based on characteristics expected of ChIP data. The results show that RN reduces the number of peaks called (Figure 4B). MOP- and FLOP-RN result in sets of peaks that contain a higher proportion of expected positives, such as replication origins, with overlaps between sets depicted by Venns (Figure 4B,C). Fewer peaks are consistently called in the FLOP-RN versus MOP-RN datasets. We think this is due to the higher expression of LexA-FLAG from the FLOP construct, resulting in more stringent competition for IP. Note that virtually all Fkh1-FLAG-controlled peaks are included in the Fkh1-MYC-controlled peak set, supporting the greater stringency of the Fkh1-FLAG set (Figure 4C). Additionally, the larger Fkh1-MYC set captures a proportionately larger number of replication origins, indicating that the larger set contains many true positives missed in the more stringent set. These conclusions are further validated below through an alternative control approach. The heatmaps largely recapitulate the results with unsynchronized cells, eliminating the enrichment of hyperChIPables and tRNAs, while maintaining enrichments for CLB2-cluster genes and Fkh-activated origins (Figure 4D).

### 2.5. Analysis of Replication Origin Binding Proteins Validate Approach for HA

To evaluate our approach with the HA epitope, we turned to two proteins we previously analyzed successfully using ChIP and ChIP-chip, the replication origin binding proteins Orc1 and Mcm4, members of the ORC (Origin Recognition Complex) and MCM (Mini-Chromosome Maintenance) complexes, respectively [11,12]. We epitope-tagged these proteins with 3xHA in the HOP strain background and performed ChIP-seq with G1-synchronized cultures of HOP, *ORC1-HA* (HOP), and *MCM4-HA* (HOP) strains. Experimental replicates show a high correlation (Appendix A). Because replication origins in yeast have been mapped and many functionally confirmed by multiple approaches, the known origins comprise a powerful set to validate our ChIP of these proteins. We used the extensive OriDB list of 829 origins categorized therein as confirmed (410), likely (216), and dubious (203) [13].

We generated chromosome plots of the results before and after RN and indicated the positions of confirmed origins on these plots as well as peaks called by MACS and generated a chart of overlap between called peaks and origins in different categories (Figure 5A,B). Venn diagrams show the overlaps between uncontrolled and RN datasets with origins and compare Orc1 versus Mcm4 overlaps with and without RN (Figure 5C and Appendix A). The peak-calling analysis reveals a substantial reduction in the number of dubious origins called relative to the other categories with the application of HOP control normalization (Figure 5B). For Orc1, a similar number of confirmed origins were called peaks in the controlled (254) and uncontrolled (268) dataset, whereas the number of “other” peaks was reduced nearly 3-fold in the RN (99) versus uncontrolled (279) set. Similarly, in the RN versus uncontrolled data, dubious origins were also reduced (from 56 to 22), while origin in the likely category were reduced intermediately to the confirmed and dubious, as expected. Thus, the application of HOP-RN to Orc1-HA increased the proportion of total called peaks accounted for by a confirmed or likely origin from 52% (356/691) to 73% (319/440), while only reducing the total number of called origins in this grouping by 10%.

For Mcm4, the overall results followed a similar pattern with the application of HOP-RN reducing calls at “other” and “dubious” loci more than “confirmed” and “likely” origin loci. For example, peak calls overlapping with confirmed origins decreased by 20% (from 247 to 197), while dubious and other calls decreased by 81% (from 402 to 75) (Figure 5B). There was a lower overall number of Mcm4 than Orc1 peaks called in HOP-RN sets, which may reflect loci that are inefficient at MCM loading [12]. We tested whether a relationship exists between CHIP signals for Orc1 and/or Mcm4, presumed to represent the binding occupancy of these proteins, and the efficiencies of individual origins, as MCM stoichiometry at replication origins has been proposed to regulate replication timing [14]. However, no significant correlation was found between origin efficiencies and Orc1 or Mcm4 ChIP signals (Appendix A).

We also generated heatmaps of hyperChIPable loci, *tRNAs*, and origins, and observed a modest but significant enrichment for hyperChIPable and *tRNAs* in *ORC1-HA* HOP-RN, but not with *MCM4-HA* HOP-RN. Enrichments may reflect the co-localization of origins with *tRNAs*, which has been previously reported [12]. Regardless, the application of RN reduces signal enrichment at hyperChIPable and *tRNA* loci, while maintaining, if not enhancing, origin signals for both ORC1-HA and MCM4-HA (Figure 5D).

The examination of the chromosome plots of the control anti-HA IPs shows the strong enrichment of centromere (*CEN*) sequences, which is also shown by a heatmap of averaged data at *CEN* sequences (Figure 5A,D). We also noticed that signals at replication origins were under-enriched in the HOP control dataset, contributing to an improvement in the signal upon RN (Figure 5A,D and Appendix A, Appendix A). Though modest, a similar hypoChIPability of origins signals was significant in the anti-MYC and anti-FLAG controls (Appendix A, Appendix A). In contrast to the anti-HA IPs, *CEN* sequences were significantly under-enriched in the anti-MYC and anti-FLAG controls (Appendix A, Appendix A).

### 2.6. Controls Enhance Analysis of Potential Hyperchipable Loci

Given the utility of our approach in detecting and eliminating hyperChIPable signals, we wondered how it would perform at a potentially hyperChIPable locus of interest such as the highly expressed *rDNA*, which has indeed been identified as hyperChIPable [5]. Each *rDNA* repeat contains a potential replication origin, and we functionally identified the *rDNA* origin(s) as Fkh-activated [9]. In the course of these studies, we also analyzed Sir2-3xFLAG(FLOP), which has been reported to bind and function within the *rDNA* to regulate gene expression and recombination [15]. Unlike the largely sequence-specific DNA binding of Orc1 and Fkh1, Sir2 is recruited to specific loci by other proteins, including Orc1 indirectly, and then spreads in cis along chromatin to deacetylate histone tails. We plotted the G1-phase ChIP-enrichment of Sir2-FLAG, Orc1-HA, Mcm4-HA, and Fkh1-MYC at the *rDNA* before and after RN (Figure 6A). With the exception of Fkh1-MYC, each of the uncontrolled ChIPs shows substantial signal enrichment across the entire *rDNA* locus with several peaks and valleys. In contrast, RN normalized data show sharp, narrow peaks at specific loci, and the substantial reduction in the signal across most of the region, with the exception of the Sir2-FLAG data, which show a combination of sharp peaks above a high baseline of enrichment across the *rDNA* region. Orc1, Mcm4, and Fkh1 peaks co-localize to the origin sequences (ARS1200-1,2), while Sir2 spreads across the region with a major peak at the I element and a secondary peak aligning with the replication fork barrier (RFB) (Figure 6A). These findings are consistent with available knowledge [16] and suggest that our controls enhance the detection of bona fide protein binding even within highly expressed loci such as the *rDNA*, which may exhibit non-specific enrichment. Inherent differences in protein binding modes may also be evident in the unique spreading of the Sir2 signal not observed for Orc1, Mcm4, and Fkh1 in these experiments. We also plotted Sir2 as a function of the distance from telomeres, highlighting other known Sir2 binding loci, such as *HML* and *HMR* (Figure 6B), and we plotted full data for chromosome III, which contain the silent loci (Figure 6C). These data show that most Sir2 enrichment occurs near telomeres, and in the vicinity of *HML* and *HMR* where Sir2 enrichment appears to be maximal between the silencer elements. Enrichment at *HML* extends to the telomere, while enrichment appears isolated from the telomere at the more telomere-distant *HMR* (Figure 6C).

### 2.7. DNA-Binding Mutant May Be Ideal Control

We tried another approach that in principle should provide the perfect control for ChIP: the expression of a DNA-binding mutant (dbm) of the target DNA-binding protein. Such a control would have the same expression level, avidity for the antibody, and similar non-specific chromatin binding properties that might contribute to a non-specific ChIP signal. The approach requires prior knowledge and/or availability of such a mutation in the target gene. As the Forkhead DNA-binding domain structure has been previously determined [17,18], it was possible for us to create a mutant allele with specific amino acid changes in the DNA sequence recognition and binding surface, that we termed *fkh1-dbm*; we confirmed the defect in DNA binding by EMSA (Appendix A). We constructed MYC- and FLAG-tagged versions of *fkh1-dbm* and replaced *FKH1* and conducted ChIP-seq in unsynchronized *FKH1-MYC*, *fkh1-dbm-MYC*, *FKH1-FLAG*, and *fkh1-dbm-FLAG* (lacking MOP or FLOP constructs). Data were processed and analyzed as above; correlation of replicates was confirmed (Appendix A).

Chromosome plots and heatmaps show the complete elimination of enrichment at specifically enriched loci, confirming the *fkh1-dbm* defect in vivo (Figure 7A and Appendix A). Furthermore, fkh1-dbm exhibited continued enrichment for hyperChIPable and *tRNA* loci. Thus, RN using *fkh1-dbm* (dbm-RN) data results in the elimination of hyperChIPable signals while retaining the enrichment of the signal at replication origins and *CLB2*-cluster genes (Figure 7A). Peak-callings were performed on the RN data and compared as above. For both IPs, data normalized by *fkh1-dbm* as opposed to MOP or FLOP yielded more peak calls, 795 versus 501 for MYC and 474 versus 286 for FLAG (Figure 7B). Most peaks in the MOP- or FLOP-controlled sets were also called in the larger dbm-RN sets, which performed similarly to MOP-RN and better than FLOP-RN at capturing Fkh-activated origins and *CLB2* cluster genes in proportion to set sizes. We conclude that the DBM control is likely a superior approach when available.

## 3. Discussion

### 3.1. Expression of a Decoy Protein to Control for Non-Specific Sequence Enrichment

Our results show that the in vivo expression of a heterologous or non-DNA-binding protein sharing an epitope tag with the experimental target protein provides an effective sensor for ChIP to identify non-specific signals for use in data normalization, and thereby can greatly enhance the quality of ChIP-seq analysis. Our approach requires data from two ChIPs (each of which should be replicated): the control strain expressing the “mock” binding protein (e.g., MOP, HOP, FLOP) and the experimental strain also expressing the protein of interest tagged with the same epitope as the control protein. These datasets are scale-normalized, and the experimental set is divided by the control set, yielding ratio-normalized data. We validated our data through a series of experiments using proteins with known targets. Our results yield new, high-quality data for Fkh1 in unsynchronized cells, and Fkh1, Orc1, Mcm4, and Sir2 in G1-synchronized cells. More broadly, our results validate an experimental control paradigm and provide constructs for working with epitope-tagged proteins in ChIP.

We chose LexA due to its natural absence from yeast along with its established use as a robust and specific DNA-binding protein to its specific target sequence [19]; it was also convenient as we already possessed DNA sequences containing elements for expression and binding to use for the control constructs. We see no reason why other proteins could not be similarly used, and they actually might function better. In fact, we show that a potentially ideal control, if available knowledge and/or resources exist, is the expression of a DNA/chromatin-binding defective version of the target protein to use for ratio normalization. Our findings strongly emphasize the value of the efforts required to implement these controls.

For proteins that are recruited to chromatin by other factors, ChIP in cells depleted of the recruitment factor(s) should provide a rigorous test of specificity. Inclusion in the expression vector of lexA DNA-binding sequences provides a positive control for the expressed LexA protein in the ChIP with potential information about the resolution of the data. In addition, these internal controls can serve as internal standards against which other loci may be measured, or to confirm alternative normalization methods.

### 3.2. Multiple Factors Contribute to Non-Specific ChIP-Signal Enrichment

Analysis of non-specific signal enrichment showed that numerous factors influence such enrichment, including the antibody used for IP, the expression of antibody target protein(s), both control and experimental, and the epitope tags. The Fkh1 analysis suggests that IPs against MYC were more specific than against FLAG; however, it should be recalled that the number of repeats for the tags varied. In addition, the cell cycle stage of analysis also impacted non-specific sequence enrichment. For example, the hyper-enrichment of *CENs* in anti-HA IPs was much enhanced in G1 phase. In contrast, *CENs* were hypo-enriched in the FLAG and HA IPs and replication origins also in all three IPs. Overall, we showed that carrying out ChIP with appropriate control strains identifies these biases in the data, enabling their effective subtraction from the data to yield more accurate results.

Our results indicate that the presence of a target protein for the antibody alters non-specific IP, probably at least in part due to competition for binding the antibody. Compared to an untagged strain, the strains expressing the heterologous epitope-tagged protein appear to more greatly enrich hyperChIPable or other non-specific loci that are also likely to be enriched in the ChIP against the protein of interest, and this appears to be key to their action here. It is possible and likely that the control protein IP enriches for additional, non-specific sequences than the target protein IP, which is why the experimental strain also expresses the control protein. In this regard, we think the dbm approach is most ideal because the dbm protein is biochemically identical to the target protein of interest, save a few amino acid changes disrupting high-avidity, sequence-specific binding. Thus, other interactions that might contribute to non-specific IP will likely be retained, allowing their subtraction, with less risk of adding new, non-specific targets of IP.

### 3.3. No Control, No Experiment

The powerful insight provided by bona fide ChIP results in identifying a protein–DNA/chromatin interaction in vivo, which has made such analysis de rigueur. However, ChIP applications, especially for new discoveries, are fraught with uncertainty regarding absent stringent attention to controls to validate results and, as we have shown, to correct for intrinsic, yet unpredictable sequence enrichment biases. Even with a previous application of untagged strain controls, we failed to correct for biased enrichment, an error we have now rectified. The obvious controls are critical to attempt, but the best controls are not always obvious or feasible. With the application of effective controls as we describe here, ChIP datasets can be filtered of contaminants and made more inherently valuable for all downstream uses.

## 4. Materials and Methods

### 4.1. Plasmid Constructions

Primer sequences and descriptions are given in Appendix A. Plasmids pADE2-MOP, pTRP1-HOP, and pADE2-FLOPv2x2 (herein otherwise referred to as pADE2-FLOP) were constructed by Gibson assembly (SGI #GA1200) and/or standard restriction endonuclease digestions and ligations (New England Biolabs, Ipswich, MA, USA), according to the manufacturer’s protocols. 13xMYC, 3xHA epitope tags, and *TEF1* promoter sequences were PCR amplified from pFA6 vectors [20]; 3xFLAG was amplified from p2L-3Flag-TRP1 (T. Tsukiyama, Seattle, WA, USA); and *ADH1* sequences and lexA protein coding and lex Operator binding sequences were amplified from pJL4 and pEC15 [21]. Partial *ADE2* and *TRP1* target sequences (from BY4741) for genome targeting and selection were designed to integrate the constructs stably into the homologous regions, bearing the auxotrophic mutations in the W303 background (i.e., *ade2-1* and *trp1-1*). His6-tagged protein expression and the construction of p405-FKH1 has been described previously [22]. Site-directed mutagenesis was used to produce pET28a-Fkh1-dbm using Quickchange Multi kit Agilent, Santa Clara, CA, USA). Plasmid sequences were confirmed by DNA sequencing (Retrogen Inc., San Diego, CA, USA).

### 4.2. Yeast Strain Constructions

Primer sequences are given in Appendix A. Genotypes of all yeast strains are given in Appendix A. ADE2-MOP/FLOP and TRP1-HOP constructs were liberated from plasmid vectors by digestion with *SacI* + *KpnI* and transformed into yeast with the selection of -ade or -trp medium as appropriate. DNA was introduced into yeast by lithium acetate transformation with an appropriate selection [23] and was confirmed by PCR. The expression of all epitope-tagged proteins was confirmed by Western blot analysis of whole cell protein extracts produced by TCA precipitation, using the following monoclonal antibodies: anti-FLAG (Sigma M2, Cat#F1804, St. Louis, MO, USA), anti-MYC (BioLegend 9E10, Cat#626802, San Diego, CA, USA), or anti-HA (Thermo-Fisher 12CA5, Cat#11583816001, Waltham, MA, USA; or BioLegend 16B12, Cat#901514).

### 4.3. Other Methods

Yeast cultures were grown at 23 °C in YEPD and harvested during logarithmic growth or after 3 h of growth in the presence of 5 nM ⍺-factor. ChIP-Seq was performed in triplicate for each strain using the antibodies listed above at 1:200 according to our protocol and analysis pipeline in [24], with the following minor alterations: chromatin shearing with the Covaris (Woburn, MA, USA) S2 sonicator (duty cycle 20%, intensity 5, cycles/burst 200, for three 30 s cycles) was performed three times with 10 min on ice between cycles. Following IP, before proceeding to library construction, the purified DNA was subjected to a second shearing step using the Covaris S2 sonicator as follows: (duty cycle 10%, intensity 4, cycles/burst 200, for four 30 s cycles). Libraries were constructed using KAPA HyperPrep Kit (Roche, Indianapolis, IN, USA).

Amplified libraries were subjected to final quality control and quantifications for high-throughput sequencing using Illumina technology (150 bp paired-end), carried out by Novogene (Davis, CA, USA). Sequencing files were demultiplex using the ‘fastq-multx’ function in ea-utils version 1.1.2-806 and then aligned to the genome using Bowtie2 version 2.3.5.1. The mapped read files were converted to BAM format using the ‘view’ function in Samtools version 1.10. Reads were then pair matched using the ‘fixmate’ function in Samtools and sorted using the ‘sort’ function in Samtools. Duplicates were removed using the ‘markdup’ function in Samtools. These BAM files were then converted to BED files using the ‘bamtobed’ function in Bedtools2 version 2.27.1. The coverage of sequence alignments was then calculated across 50 bp windows using the ‘bamtobed’ function in Bedtools2. Sequencing data are available at GEO (GSE230475). Peaks were called by MACS version 2.2.7.1 using the BAM files following the removal of duplicates, genome size of 1.24 × 10^7^, no model, extension size of 100, a q-value cutoff of 5 × 10^−2^, and, when indicated, a control file.

To generate plots and perform statistical analysis, each binned coverage file was processed with a Matlab script that generates RN values for each experiment (see Appendix A). The Matlab script RN_normalization was used to import the numbered. bed files and smooth across a set window (e.g., 500 bp). Then, data were scale-normalized using a set percentile as the background (e.g., 20%). Next, triplicates were averaged, and then experimental sets were divided by the appropriate control. The list of hyperChIPable sequences is from [5]; the list of origins is from oriDB (cerevisiae.oridb.org (accessed on 16 March 2021)); and the list of Fkh-activated origins is from [9]. *S. cerevisiae* genome sequence (last modified date: 25 October 2019, Time: 18:55:47.000Z) and element assignments (e.g., *tRNAs*, *CENs*) are from the Saccharomyces Genome Database (yeastgenome.org (accessed on 17 March 2021)). Origin efficiencies are from [25].

Electrophoretic mobility shift analysis (EMSA) was performed as follows: *TEM1* DNA probe was produced by annealing oligonucleotide primers (Appendix A). On ice, 10 ng of DNA probe was combined with protein in a 20 μL total volume in 20 mM Tris-HCl pH 7.9, 50 mM KCl, 5 mM MgCl_2_, 3 mM DTT, 0.1 mg/mL BSA, 10% (*v*/*v*) glycerol. Binding reactions were incubated on ice for 15 min, followed by incubation at room temperature (22 °C) for 15 min. Samples were loaded onto a 10% (*w*/*v*) (21:1, acrylamide/bis-acrylamide) polyacrylamide gel and separated in a 0.5× TBE buffer at 120 V for 90 min in a 4 °C environmental chamber. The gel was stained in 0.5× TBE, 0.2 nM SYBR Green I (Molecular Probes, Eugene, OR, USA) for 10 min at 22 °C with gentle mixing. The gel was de-stained by incubating in H_2_O for 10 min at 22 °C with gentle mixing and repeating. The gel image was captured using a BioRad (Berkeley, CA, USA) FX scanner.

## Figures and Tables

**Figure 1 ijms-24-09271-f001:**
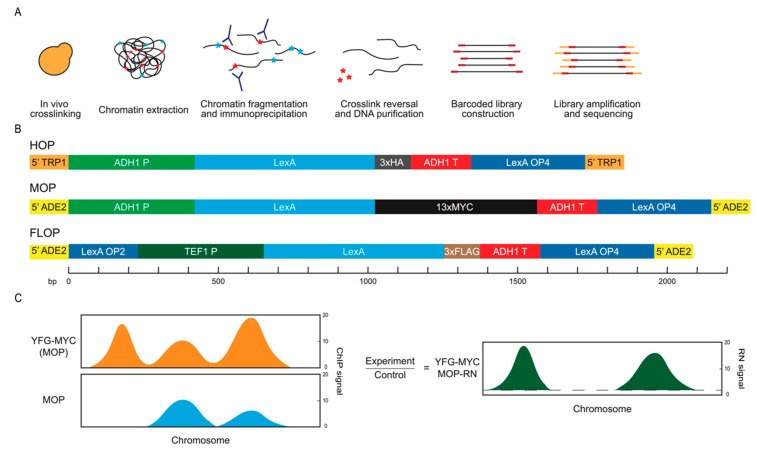
Control constructs and analysis scheme. (**A**) Standard ChIP-seq procedure performed here. Red stars represent the protein of interest and blue stars represent other proteins. Red rectangles represent unique barcode sequences and orange rectangles indicate universal indexes. (**B**) Schematic representation of DNA elements comprising the different control constructs, not showing the plasmid vectors. The *ADE2* and *TRP1* sequences target integration of the constructs as shown upstream of the promoter regions of these loci; P and T represent promoter and terminator sequences and LexA-OP2/4 represent two or four LexA binding sequences. (**C**) Schematic representation of ratio normalization (RN) to remove background signal.

**Figure 2 ijms-24-09271-f002:**
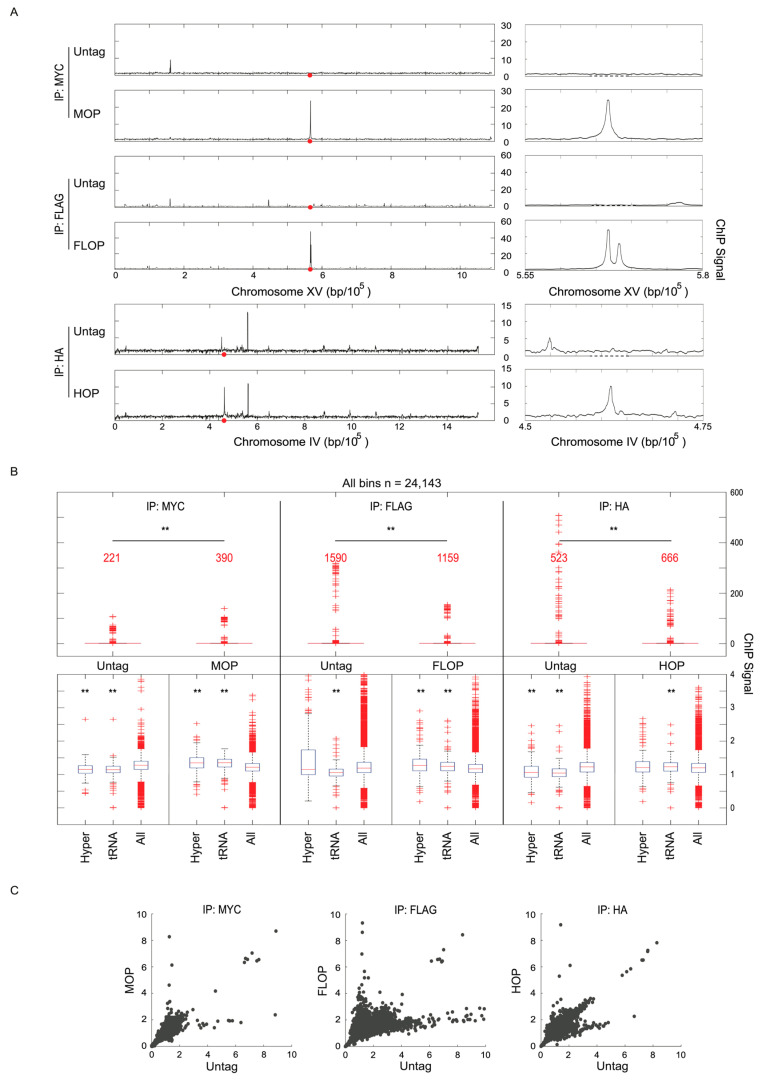
Enrichment of non-specific ChIP signals in control IPs. Strains MPy105 (MOP), MPy35 (FLOP), and MPy39 (HOP) were grown to log-phase, harvested, and analyzed using ChIP-seq. (**A**) Plots of ChIP signals across chromosomes XV and IV, which harbor the MOP, FLOP, and HOP loci, respectively indicated by red dots; panels to the right show zoomed-in view. (**B**) Distribution boxplots of ChIP signals (500 bp bins) across the whole genome and for specific sets of loci; results of Mann–Whitney tests of difference in distributions indicated by asterisks ** *p* < 0.01. Outliers are indicated with + and the number of positive outliers is indicated for each in red. (**C**) Two-dimensional scatter plots of ChIP signals for all bins for Untag versus MOP, FLOP, or HOP for each set of IPs.

**Figure 3 ijms-24-09271-f003:**
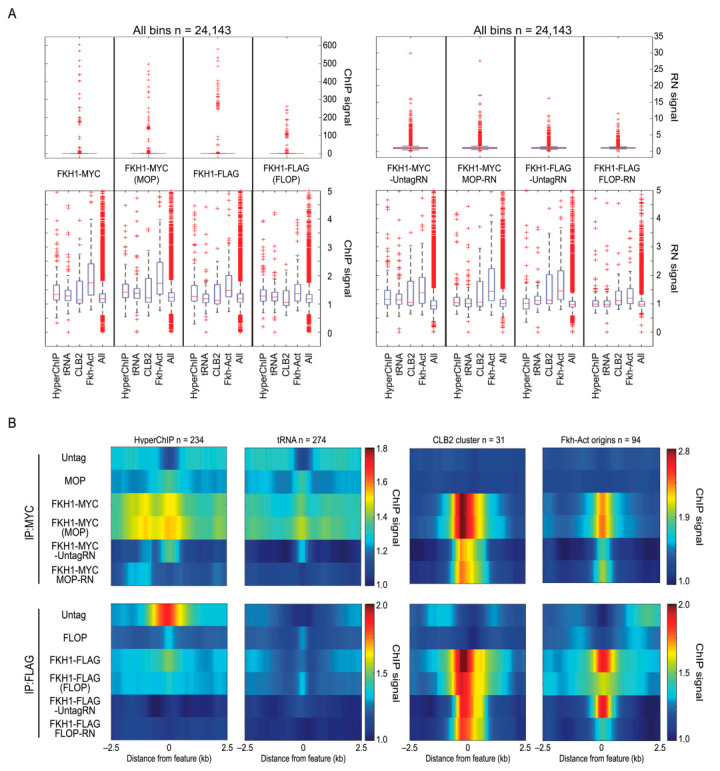
Ratio normalization using control reduces non-specific enrichment. Strains SSy161 (*WT*), MPy105 (MOP), MPy166 (*FKH1-9xMYC*), MPy108 (*FKH1-9xMYC*(MOP)), MPy35 (*FLOP)*, OAy1100 (*FKH1-3xFLAG*), and MPy55 (*FKH1-3xFLAG(*FLOP)) were grown to log-phase, harvested, and analyzed using ChIP-seq. (**A**) Distribution boxplots of ChIP signals (500 bp bins) across the whole genome and for specific loci; RN is the ratio normalized signal. (**B**) Heatmaps of averaged ChIP signal across 5 kb regions centered on the indicated features.

**Figure 4 ijms-24-09271-f004:**
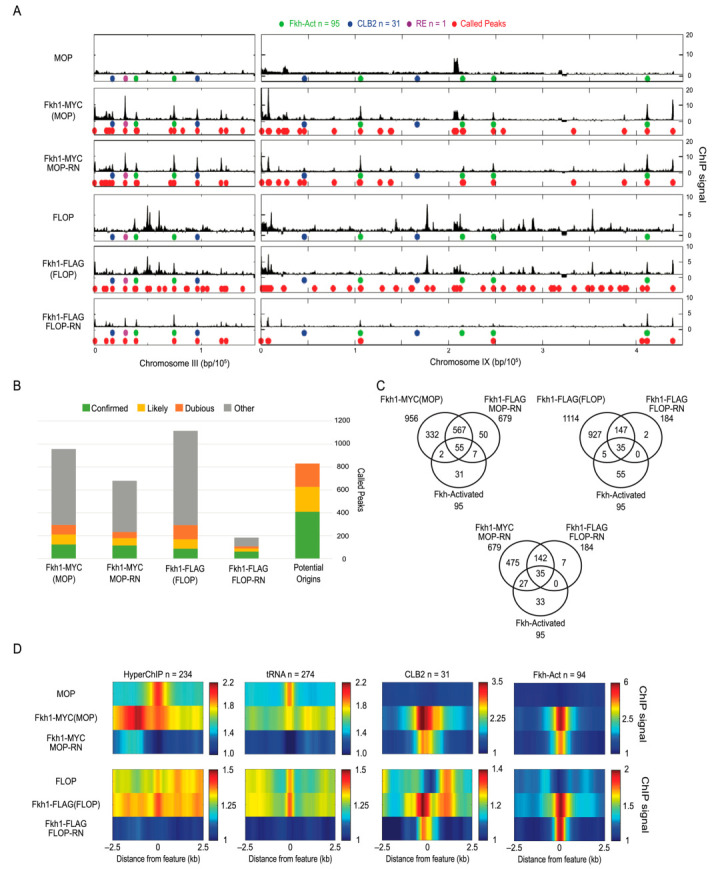
Improved detection of Fkh1 binding loci. Strains MPy105 (MOP), MPy108 (*FKH1-9xMYC*(MOP)), MPy35 (FLOP), and MPy55 (*FKH1-3xFLAG*(FLOP)) were synchronized in G1 phase, harvested, and analyzed using ChIP-seq. (**A**) Plots of ChIP signal across chromosomes III-L and IX with potential Fkh1 binding sites indicated as colored circles on one track and called peaks indicated on the lower track. (**B**) Stack graphs of peak calls overlapping with potential origins according to their categorization in oriDB or with other loci presumed not to contain replication origins. (**C**) Venn diagrams showing overlap of called peaks with origin sets and the MYC/MOP versus FLAG/FLOP sets. (**D**) Heatmaps of averaged ChIP signal across 5 kb regions centered on the indicated features.

**Figure 5 ijms-24-09271-f005:**
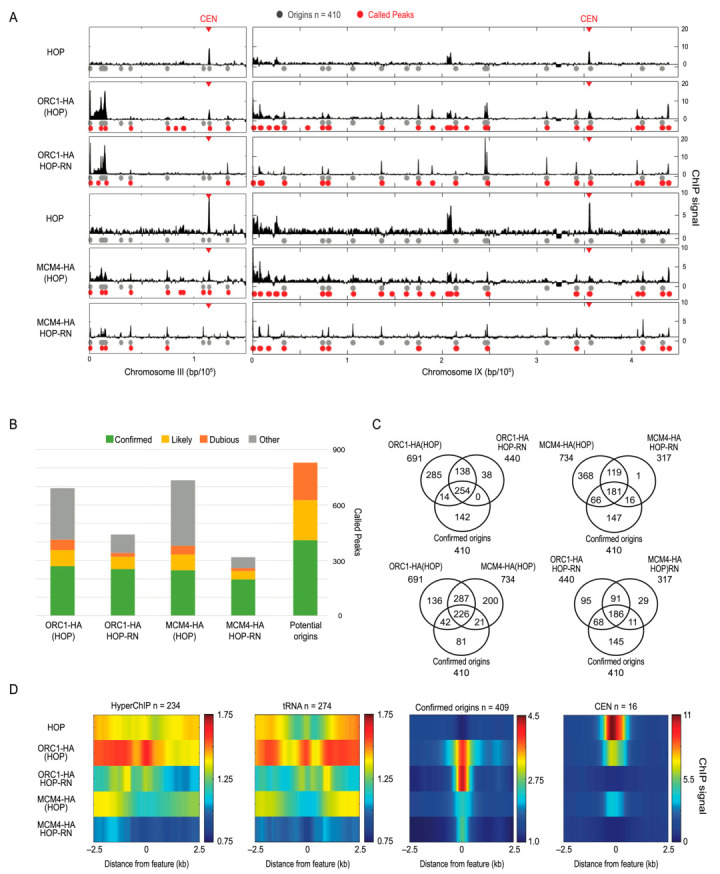
Improved accuracy in detection of ORC and MCM binding loci. Strains MPy39 (HOP), MPy199 (*ORC1-3xHA*(HOP)), and MPy102 (*MCM4-3xHA*(HOP)) were synchronized in G1 phase, harvested, and analyzed using ChIP-seq. (**A**) Plots of ChIP signal across chromosomes III-L and IX with potential origins indicated as gray circles on one track and called peaks indicated on the lower track in red and *CEN* indicated by arrowhead. (**B**) Stack graphs of peak calls overlapping with potential origins according to their categorization in oriDB or with other loci presumed not to contain replication origins. (**C**) Venn diagrams showing origin overlap of peaks called using RN and uncontrolled datasets against confirmed origins. (**D**) Heatmaps of averaged ChIP signal across 5 kb regions centered on the indicated features.

**Figure 6 ijms-24-09271-f006:**
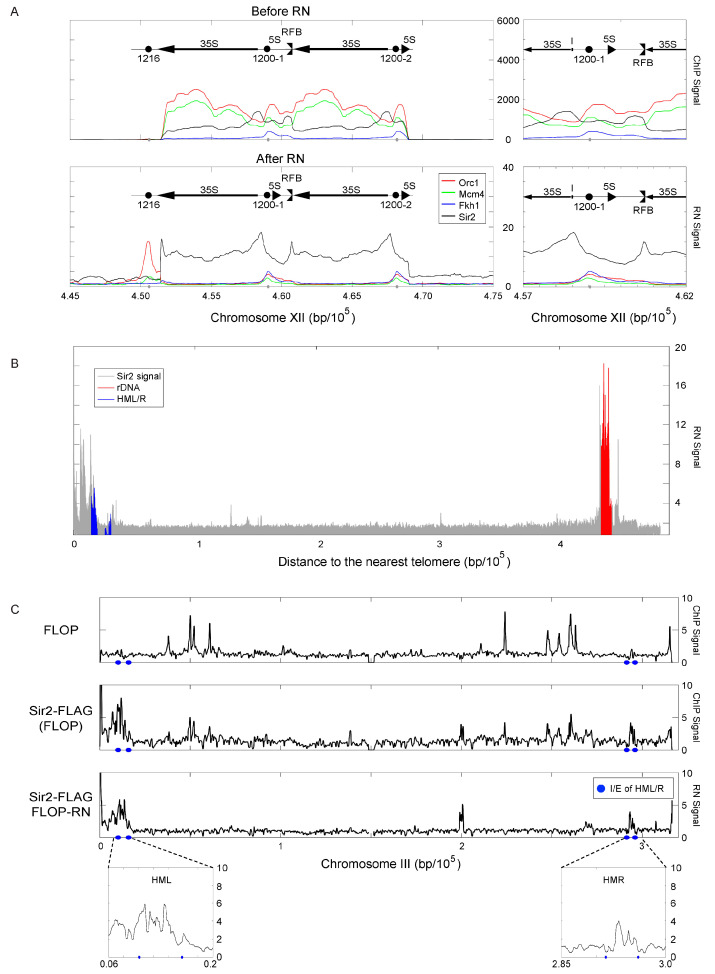
Sir2 detection within hyperChIPable *rDNA* locus and genome-wide. (**A**) Plots of ChIP signals overlaid across the *rDNA* region for G1-synchronized strains MPy108 (*FKH1-9xMYC(*MOP)), MPy199 (*ORC1-HA*(HOP)), MPy102 (*MCM4-3xHA*(HOP)), and YHy29 (*SIR2-3xFLAG*(FLOP)) before and after RN. (**B**) Plot of ChIP signals for all bins according to distance from the nearest telomere; bins mapping to the indicated loci are highlighted. (**C**) Plots of ChIP signals across chromosome III, with expanded view of *HML* and *HMR* silencer regions and silencer elements (E and I) indicated; correlation of replicates shown in Appendix A. Because the reference genome is *MATα*, while the analysis strain is *MATa*, we deleted ⍺ gene sequences at the *MAT* locus in the reference sequence to properly map all ⍺ gene sequence reads to *HMLα*.

**Figure 7 ijms-24-09271-f007:**
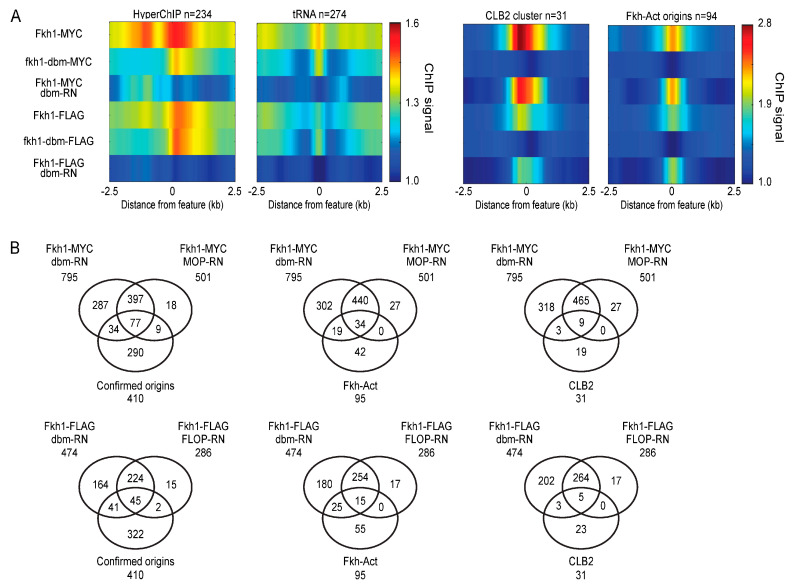
DNA-binding mutant control may be ideal. Strains MPy166 (*FKH1-9xMYC*), MPy169 (*fkh1-dbm-9xMYC*), MPy172 (*fkh1-dbm-3xFLAG*), and OAy1100 (*FKH1-3xFLAG*) were grown to log-phase, harvested, and analyzed using ChIP-seq. (**A**) Heatmaps of averaged ChIP signal across 5 kb regions centered on the indicated features. (**B**) Venn diagrams showing overlap of called peaks with origin and *CLB2* cluster sets comparing dbm-controlled against MOP- and FLOP-controlled sets analyzed in Figure 3.

## Data Availability

Sequencing data is available at GEO (GSE230475).

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
