# Peer review of "Broadly Applicable Control Approaches Improve Accuracy of ChIP-Seq Data"

_ijms, 2023, doi:10.3390/ijms24119271_

Round 1

Reviewer 1 Report

The manuscript by Petrie et al. presents a novel and broadly applicable strategy for control protein selection that allows to dramatically improve accuracy of ChIP-seq data. ChIP-seq is a powerful method for identifying genome-wide DNA binding sites for a protein of interest, however false-positive enrichment remains a problem. The authors of this study demonstrate that using a control non-DNA binding protein with the same epitope tag as a protein of interest results in better identification of non-specific binding sites and in robust improvement of data quality. In addition, the authors demonstrate that a version of a protein of interest with mutations in DNA-binding interface presents an ideal control for ChIP-seq experiments.

Overall, the study is well performed and described and can be published in the current form.

Author Response

We thank the reviewer for their effort and positive comments. 

Reviewer 2 Report

In this manuscript, the authors addressed the issue of non-specific enrichment in TF ChIP-Seq and proposed two approaches. It was demonstrated that the epitope tag only control and the DNA binding-defective control were powerful in improving the accuracy of detecting true DNA binding signals. Since these two controls are often omitted, the findings in this manuscript would benefit the experimental design and assessment of ChIP experiment quality in different systems. It is suggested to add the description of RN in the legend of Figure 3, since it is not a common unit but a ratio normalization according to the authors' approach.

Author Response

We thank the reviewer for their effort and positive comments.  We have made the recommended edit to describe “RN” in the Figure 3 legend.

Reviewer 3 Report

In this paper, the authors describe a method for eliminating false positives in the ChIP assay. This technique is expected to help reduce discrepancies that may arise in the interpretation of the results of this experiment by comparing ChIp data across multiple plasmid constructs.

Minor comments:

1. Please provide a summary of the tools (e.g., R package) or code used to allow other researchers to reproduce the data analysis process after making peak calls in MACS.

2. Is the number of repeats of each tag used in the experiment related to the appearance of false positives? If you have data to determine this, please discuss.

3. There are several paragraphs that have been changed in the middle of a sentence. Please correct them to the correct form.

Author Response

Thank you for your time and effort; our responses to “Minor Comments” (in red) follow:

  1. Please provide a summary of the tools (e.g., R package) or code used to allow other researchers to reproduce the data analysis process after making peak calls in MACS. This has been added in the Methods and there is now an additional Table S5 with Matlab code.
  2. Is the number of repeats of each tag used in the experiment related to the appearance of false positives? If you have data to determine this, please discuss. We do not have the data that would be required to answer this properly, which would involve varying the number of tags systematically with each epitope. Our data only allows comparison of 3HA versus 3FLAG versus 13MYC, which can be seen in Fig. 2B and C. We determined the number of positive outliers in those datasets as a measure of false-positives:

Untag (IP:FLAG) = 1590

Untag (IP:MYC) = 221

Untag (IP:HA) = 523

FLOP = 1159

MOP = 390

HOP = 666

We have now added these data to Figure 2B; however, we do not wish to make any conclusions based on this very limited data.  If anything these data suggest the number of outliers is linked to the antibody/immunoprecipitation.

  1. There are several paragraphs that have been changed in the middle of a sentence. Please correct them to the correct form. Thank you. We have done so.

Reviewer 4 Report

This paper deals with the important issue of control experiments for ChIP-seq. The authors have carried out a rather extensive study. Yeast strains expressing fusion proteins lexA and various epitopes were obtained and used as controls for yeast Fkh1, Orc1, Mcm4, and Sir2 protein binding data. Fkh1 mutants with impaired DNA binding were also obtained. ChIP-seq experiments were performed, the data were processed, and conclusions about the nature of nonspecific signals that are present in experiments of this type were suggested.

The publication contains a lot of important practical observations related to the ChIP-seq technique. The control approach proposed by the authors is undoubtedly worthy of attention. Due to the widespread use of the ChIP method, it is important to draw the attention of the scientific community to the controls. Unfortunately, a number of publications ignore this issue. In this respect, the presented publication can be a great reference for other researchers.

There are the following questions and suggestions for the paper:

1) Are there NLSs in the control fusion proteins?

2) Since ChIP-seq data is often controlled using ChIP-qPCR at individual loci, it would be interesting to know what the false positive signals described in the publication look like when tested with ChIP-qPCR? Are they different from true positive signals? In other words, are the ChIP-seq and ChIP-qPCR data different in the authors' experimental models? Would the approach proposed by the authors also be useful for ChIP-qPCR experiments? Additional ChIP-qPCR experiments would help to broaden the interpretation of the data obtained.

3) Since the paper is mainly of a methodological character, the Materials and Methods section should be extended. In particular, the ChIP-seq protocol used would be helpful to give as much detail as possible rather than as references. There is a question about the source of the antibodies used - were they obtained in the lab or purchased? It is better to specify all methodological information in as much detail as possible.

Overall, the publication can be recommended for publication, taking into account the comments raised.

Author Response

We thank the reviewer for their effort and positive comments.  Our responses follow:

1) Are there NLSs in the control fusion proteins?

Functionally, the evidence strongly suggests so, given the very robust ChIP signal at the lex operator sites (Fig. 2).  LexA contains a match to the classic bipartite NLS sequence (K-K/R-X-K/R), however, we have not tested nor are we aware of tests of the importance of this or any other putative NLS in LexA.

2) Since ChIP-seq data is often controlled using ChIP-qPCR at individual loci, it would be interesting to know what the false positive signals described in the publication look like when tested with ChIP-qPCR? Are they different from true positive signals? In other words, are the ChIP-seq and ChIP-qPCR data different in the authors' experimental models? Would the approach proposed by the authors also be useful for ChIP-qPCR experiments? Additional ChIP-qPCR experiments would help to broaden the interpretation of the data obtained.

We appreciate the reviewer’s curiosity, and we certainly think there’s no basis for thinking these controls would not perform similarly for ChIP-qPCR as the methods are fundamentally similar and reliant on PCR amplification within quantitative ranges.  We think that ChIP-seq is fundamentally superior due to the complete sequence coverage, which is unattainable by qPCR.  How many false-positive and false-negative loci would need to be tested-keeping in mind that multiple probes per site would be needed for a robust comparison?  This would be time and cost prohibitive, and beyond the scope this effort; unfortunately, we don’t have the resources.  With current ease of library preparations and low sequencing costs, we think qPCR rarely makes sense for a difficult analysis like ChIP, where seeing the genome is highly revealing of the quality of the data.

3) Since the paper is mainly of a methodological character, the Materials and Methods section should be extended. In particular, the ChIP-seq protocol used would be helpful to give as much detail as possible rather than as references. There is a question about the source of the antibodies used - were they obtained in the lab or purchased? It is better to specify all methodological information in as much detail as possible.

The paper reference for the ChIP-seq method is highly detailed chapter (13 pages) in Methods in Molecular Biology, which certainly contains more detail than can be feasibly reproduced here.  It is in fact our bench protocol.  Nevertheless, we have expanded on any differences from the previous protocol, which are minimal, and we provide additional source information on the antibodies.

Round 2

Reviewer 4 Report

Unfortunately, the authors provided a superficial answer instead of a thorough analysis of the questions posed.

1) Do the authors consider the localization of the control protein in the nucleus to be important? Most likely, it is important to create the adequate control that the authors claim. If this is the case, immunostaining of the cells and verification of the localization of the proteins in the nuclei of the cells should be performed.

2) When discussing sequencing data, the authors regard ChIP-seq as an absolute truth. Scientific papers of a good enough level should focus on confirming the results obtained using several techniques.

NGS technology can introduce distortions. In the field of the ChIP method, this issue has not been sufficiently studied. This issue has been studied in more detail in the field of RNA-seq, which is more widely used. An important observation is the requirement of similarity of the repertoire of RNA molecules in the compared samples (doi: 10.1261/rna.074922.120). Direct comparison of NGS data in samples that differ significantly in the spectrum of the molecules represented can give incorrect results.

In the presented work, the emphasis is precisely on a new method for comparing a positive signal (the transcription factor of interest) and a negative control (a protein with no potential binding sites in the genome). Generally speaking, we can expect that the representation of different genome fragments will differ significantly in these samples. In the case of the negative control, only random fragments will be present, while in the experimental sample, specific loci will be significantly enriched. Therefore, libraries can differ significantly in the spectrum of DNA fragments.

The ChIP-qPCR method gives a direct estimate of the representation of a particular DNA locus in the sample. The authors can test for 2 sites each, which they consider negative, in the control lines obtained and in the experimental sample. This will require 4 primers, standard reagents and a few days. Any modern laboratory can help them with PCR.

3) The authors listed the antibody manufacturer, but were unable to add catalog numbers. This is also important for reproducible results.

Author Response

We do not consider our response to be superficial. 1) the protein has an NLS by definition and the data support that. 2) We did not refer to absolute truths and are sorry that the reviewer is unhappy that we don't consider expanding the study to include qPCR analysis imperative.  We made no reference to that technology in the paper; our claims and conclusions are more limited.  Our controls are for ChIP-seq; it's in the title. 

We can provide catalog numbers, but this does not guarantee that the same supplier or batch of antibody is available.  An internet search finds the limited choices available, but no guarantees of anything.

Round 3

Reviewer 4 Report

The authors provided some of the requested information.

Overall, the publication contains valuable information for experts in ChIP-seq analysis.